# Necroptosis-Related Gene Signature Predicts Prognosis in Patients with Advanced Ovarian Cancer

**DOI:** 10.3390/cancers17020271

**Published:** 2025-01-15

**Authors:** Mingjun Zheng, Mirjana Kessler, Udo Jeschke, Juliane Reichenbach, Bastian Czogalla, Simon Keckstein, Lennard Schroeder, Alexander Burges, Sven Mahner, Fabian Trillsch, Till Kaltofen

**Affiliations:** 1Department of Obstetrics and Gynecology, University Hospital, LMU Munich, Marchioninistrasse 15, 81377 Munich, Germany; mingjunzheng@hotmail.com (M.Z.); mirjana.kessler@med.uni-muenchen.de (M.K.); julian.reichenbach@med.uni-muenchen.de (J.R.); bastian.czogalla@med.uni-muenchen.de (B.C.); simon.keckstein@med.uni-muenchen.de (S.K.); lennard.schroeder@med.uni-muenchen.de (L.S.); alexander.burges@med.uni-muenchen.de (A.B.); sven.mahner@med.uni-muenchen.de (S.M.); fabian.trillsch@med.uni-muenchen.de (F.T.); 2Department of Gynaecology and Obstetrics, Shengjing Hospital, China Medical University, Sanhao Street 36, Shenyang 110055, China; 3Gynecology, Faculty of Medicine, University of Augsburg, Stenglinstrasse 2, 86156 Augsburg, Germany; udo.jeschke@uk-augsburg.de; 4Department for Surgery, University Hospital Regensburg, Franz-Josef-Strauss-Allee 11, 93053 Regensburg, Germany

**Keywords:** genes, necroptosis, ovarian neoplasms, RIPK3 protein, human

## Abstract

We established and validated a risk score based on a 10-gene signature from the expression of necroptosis-associated genes in advanced ovarian cancer. The patient’s prognosis, expressed as overall survival, can thus be predicted. Additionally, the risk score is able to predict the response to a programmed death ligand 1 blockade treatment in selected cancer patients. Herewith, we offer a possibility to discuss therapy from a risk stratification point of view and maybe provide an idea on how to guide one through the use of immunotherapy in solid malignancies.

## 1. Introduction

Necroptosis is a type of programmed cell death that differs from traditional apoptosis. The conventional apoptotic pathway needs caspase activation and is blocked in the absence of caspases or if the activity of caspases is suppressed, leading to the activation of necroptosis as a substitutive cell death pathway. Necroptosis is not only a process controlled by a specific molecular cascade but also characterized by cell and organelle swelling leading to cell lysis. Therefore, necroptosis has the characteristics of both apoptosis and necrosis [1,2]. It plays a dual role in malignant tumors [3,4,5,6,7]. On the one hand, the inflammatory environment induced by necroptosis can inhibit proliferation, migration and the invasion of tumor cells [4]. Further, necroptosis can inhibit tumor development by promoting the maturation of dendritic cells and the cross-activation of CD8+ T cells in the tumor microenvironment [5]. On the other hand, there are also a few studies suggesting an opposite tumor-promoting role in cancer as well [6,8].

Epithelial ovarian cancer (EOC) is the second most common cause of gynecologic cancer deaths worldwide, which seriously threatens the health of women. A lack of sufficient early detection methods often results in the disease only being recognized at advanced Fédération Internationale de Gynécologie et d’Obstétrique (FIGO) stages III/IV. In combination with the frequent problem of chemotherapy resistance, its 5-year overall survival (OS) rate remains poor at around 30–50% [9,10]. Apoptotic cell death regulation is frequently deregulated in solid malignancies. Instead, the necroptotic pathway seems to be overactivated [11], indicating that support of necroptosis could be an effective strategy to prevent the occurrence of chemotherapy resistance in EOC. Receptor-interacting serine/threonine-protein kinase (RIPK) 3 is the key component of a complex with RIPK1 and the mixed-lineage kinase domain-like pseudokinase (MLKL), which is called necrosome [12]. The canonical function of the RIPK3 signal is to stimulate MLKL activation and to trigger herewith necroptosis [13]. Besides, RIPK3 is related to mitochondrial metabolism, oxidative stress, autophagy and cell proliferation [7,14]. However, the prognostic role of necroptosis activation and RIPK3 in EOC remains unclear.

There are already different studies suggesting mechanisms for a protective role of necroptosis in EOC. For example, receptor-interacting protein-1 promotes the proliferation of EOC cells by bypassing the G2/M checkpoint and mediating the cisplatin-induced necroptosis pathway in human EOC cells [15]. Further, progesterone can prevent the occurrence of advanced serous EOC by inducing the necroptosis of p53-deficient oviduct epithelial cells [16].

This study aimed to develop a necroptosis-related gene signature and nomogram of EOC patients away from hypothesis-driven research evaluating the impact of necroptosis. Our tool might be used to predict patients’ prognosis and maybe support clinical decision making.

## 2. Materials and Methods

### 2.1. Data Source and Pre-Processing

The publicly available gene expression (RNA sequencing) and clinical data of patients with EOC were extracted from The Cancer Genome Atlas (TCGA) and the Gene Expression Omnibus (GEO). TCGA data were downloaded from the University of California Santa Cruz Xena. From the GEO database, we retrieved the GEO series 140082 (GSE140082) cohort. The according annotation file for this cohort’s RNA sequencing probes was downloaded from the chip platform GPL14951, which is used in the GSE140082 cohort.

EOC data obtained from a TCGA cohort, called TCGA-OV, and the GSE140082 cohort were processed as follows: (1) samples without clinical follow-up data were excluded; (2) samples with no survival status, an OS of <0, or unknown survival time were excluded; (3) probes were converted to a gene symbol; (4) probes corresponding to several genes were excluded; (5) the expression of several gene symbols was evaluated based on the median value; and (6) samples of FIGO stage III/IV were included and I/II were excluded. After pre-processing the downloaded data, the TCGA-OV and GSE140082 datasets were found to contain 347 and 328 EOC tissue samples, respectively (Table 1).

To test the prognostic potential of relevant genes besides *RIPK3,* we used the necroptosis gene set M24779.gmt, containing *RIPK1*, *RIPK3*, *MLKL* and five other necroptosis-associated genes (*FADD*, *FAS*, *FASLG*, *TLR3* and *TNF*). It was retrieved from the Gene Set Enrichment Analysis (GSEA) database. Additionally, based on previous studies on necroptosis, 58 necroptosis-associated genes were added to this pool (Appendix A).

### 2.2. Clustering of Necroptosis-Associated Genes and Construction of a Prognostic Model

The *ConsensusClusterPlus* package in R (version 4.3.3, The R Foundation for Statistical Computing, Vienna, Austria) with 1000 iterations and an 80% resampling rate was used to identify different molecular subtypes of EOC and define the biological properties of necroptosis-associated genes in EOC [17].

Samples in the GSE140082 dataset were divided into training and testing sets. They were randomly grouped 100 times in advance to avoid the impact of random allocation variance on the stability of the successive model. The following two criteria were used to determine an appropriate division of both sets: (1) age, FIGO stage, grading and follow-up duration of patients were identical in both groups, and (2) after random clustering of the gene expression profile datasets, both groups had a similar sample size. The two cohorts were analyzed using the chi-square test, and no significant differences (*p* < 0.05) were found, suggesting that the grouping was appropriate (Table 2).

A univariate Cox proportional hazards regression model was established based on data from the training dataset. Herewith, *p*-values < 0.05 indicated necroptosis-associated genes that are relevant to our model in terms of OS. To reduce the number of genes accounting for our risk model, we used the least absolute shrinkage and selection operator (LASSO). It is a type of compression algorithm in which a few coefficients are condensed, and others are set to zero to create a more refined model by establishing a penalty function. Therefore, the multicollinearity issue in regression analysis is resolved while retaining the advantage of subset contraction. A necroptosis-associated gene signature was developed after identifying variables via LASSO–Cox regression analysis using the *glmnet* package in R [18]. Herewith, a risk score (RS) was calculated.

### 2.3. Estimation of Cell Infiltration in the Tumor Immune Microenvironment (TIME)

The relative abundance of each cell infiltrating the TIME of EOC was calculated using the single-sample GSEA tool. A gene set identifying multiple types of infiltrating immune cells was derived from a study conducted by Charoentong and colleagues [19]. This set comprised different human immune cell subtypes: dendritic cells, activated CD8+ T cells, natural killer (NK) cells, macrophages and regulatory T cells. The relative infiltration of these cells in each sample was represented by the enrichment score calculated using a single-sample GSEA as well. We further used the Estimation of STromal and Immune cells in MAlignant Tumor tissues using Expression data (ESTIMATE) algorithm via the *ESTIMATE* package in R to determine the immune score, stromal score, tumor purity and ESTIMATE score of each patient.

### 2.4. Prediction of Immunotherapeutic Efficacy

The *IMvigor210CoreBiologies* package was used to obtain clinical data and biological information for the immunotherapy response cohort IMvigor210 [20]. The gene expression profiles were transformed from counts to transcripts per million format and subjected to a log2 transformation. This cohort contains 348 samples from urothelial carcinoma patients who received a programmed death ligand 1 (PD-L1) blocking therapy. In the absence of an adequate cohort with data from EOC patients or at least gynecologic patients, we finally identified IMvigor210 as one of the most closely related ones. The following categories were assigned based on the response of patients to PD-L1 blocking treatment: progressive disease (PD), stable disease (SD), partial response (PR) and complete response (CR). Patients with CR/PR were seen as responders to immunotherapy, whereas those with SD/PD did not respond to treatment. Receiver operating characteristic (ROC) curves were drawn using the *pROC* package. In the interpretation and visualization of the four response categories (PD, SD, PR and CR), we used the results of an analysis of variance and visualized the correlation matrix with the *Corrplot* package.

The deconvolution algorithm CIBERSORT was used to analyze the enrichment levels of 22 cell types of the TIME based on the gene expression matrix of the IMvigor210 cohort. Additionally, we used the Microenvironment Cell Populations (MCPs) counter to evaluate the infiltration of ten types of immune cells based on the gene expression profile with the *MCPcounter* package.

### 2.5. Patient Selection and Ethical Approval for Immunohistochemistry

Samples were collected from 155 patients with EOC (serous [*n* = 109], clear cell [*n* = 12], endometrioid [*n* = 21] and mucinous [*n* = 13]) who underwent radical cytoreductive surgery in the Department of Obstetrics and Gynecology at LMU Munich between 1990 and 2002. A specialized gynecologic pathologist performed histopathological diagnoses of EOC including tumor stage according to FIGO and grading. All patients, except those with low-grade FIGO stage IA, underwent platinum-based adjuvant treatment. The Munich Cancer Registry, patient files and postoperative care schedules were used to obtain patient data including information regarding relapse and mortality.

This analysis was performed in accordance with the 1964 Declaration of Helsinki (last revised in 2013) and received approval from the ethics committee of LMU Munich (reference number 138/03). All participants provided written informed consent. Statistical analysis, assessment of samples and clinical characteristics were kept anonymous.

### 2.6. Tissue Microarray and Immunohistochemistry Analysis

For the preparation of the tissue microarray, representative sections of paraffin-embedded tumor biopsy samples were cut, with a diameter of 0.6 mm, and placed into a paraffin block (30 × 20 × 10 mm) using a microtissue arrayer (Beecher Instruments, Sun Prairie, WI, USA). A total of 465 tissue microarrays were obtained by performing three biopsies on every tumor sample. Thereafter, tissues were cut into 5 µm thick sections and placed onto microscope slides. A hematoxylin and eosin staining was performed next to establish whether there is proper tumor tissue for analysis.

In the immunohistochemistry, we used a pressure cooker heating and the ZytoChem-Plus HRP Polymer-Kit (Zytomed Systems, Berlin, Germany). It includes 3,3′-diaminobenzidine as a chromogenic substrate, which is described in previous studies [21,22,23]. The primary antibody in this experiment was the polyclonal rabbit anti-RIPK3 IgG (product number HPA055087, Merck, Darmstadt, Germany) diluted 1:800. All samples were analyzed, imaged and scored using the AxioCam digital camera system combined with the AxioScope microscope (Carl Zeiss, Jena, Germany) and the AxioVision software (version 4.9.1, Carl Zeiss).

Controls were placed on every tissue microarray. Placenta tissue was used as a positive control and colon tissue was used as a negative control by replacing the primary antibody with a specific isotype control antibody (product number HK408-5R, BioGenex, Fremont, CA, USA).

For a semi-quantitative analysis of the immunostainings, we used the well-known immunoreactivity score (IRS). It is calculated by multiplying the proportion of positive cells (0: none, 1: <10%, 2: 10–50%, 3: 51–80% and 4: >80% positive cells) by the staining intensity (0: absent, 1: weak, 2: moderate and 3: strong). The IRS was independently assessed by two experienced examiners to ensure consistency and reproducibility.

### 2.7. Statistical Analyses

Statistical analyses were performed using R software.

Survival curves were plotted with the Kaplan–Meier (KM) method, and the logrank test was used to compare differences between groups. To ascertain the independent prognostic value of the RS, univariate and multivariate Cox regression models were taken for integrated analysis of RSs and additional clinical variables. ROC curves were used to estimate the ability of the developed risk model to predict the 1-, 2- and 3-year OS rates. *p*-values of <0.05 were considered statistically significant.

To build the nomogram, based on the degree of influence of each component on the OS (the size of a regression coefficient), each influencing factor was allocated with a score. These scores are then added to derive a final score. Subsequently, the predicted value of each outcome is determined based on the functional conversion correlation between the overall score and the likelihood that the outcome event will occur [24]. In our study, significant clinical factors determined via multivariate analysis were integrated as influencing factors. To verify the predictive accuracy of the nomogram, a calibration curve was simultaneously generated.

## 3. Results

### 3.1. Identification of Three Molecular Subtypes Based on Necroptosis-Associated Genes

The expression of 66 necroptosis-associated genes in each tumor sample in the GSE140082 dataset was analyzed, and consistent clustering was performed. As shown in Figure 1a,b, three molecular subtypes (k = 3) were identified: clusters A, B and C. This optimal number of clusters was determined by analyzing cumulative distribution function (CDF) curves and delta area plots. The value of k = 3 was chosen based on the point of maximum stability in cluster separation. A KM survival analysis revealed that the OS of cluster C was the shortest, while cluster B had the longest OS (Figure 1c).

The ESTIMATE algorithm was used to assess the immune, stromal and ESTIMATE scores. By analyzing the expression of specific gene signatures, the proportion of immune and stromal cells or both in the TIME can be estimated with these scores to infer the tumor’s biological composition and its interaction with surrounding tissues. In addition, the MCP counter evaluated the infiltration of ten types of immune cells. The results of both analyses showed that mostly the score was the lowest in cluster C compared to clusters A and B (Figure 1d,e). Furthermore, clustering heatmaps of immune, stromal and ESTIMATE score across different clinical features and groups (Figure 1f) demonstrated that the poor clinical prognosis of cluster C was related to a low degree of immune infiltration.

### 3.2. Construction of a 10-Gene Risk Model

By screening the 66 necroptosis-associated genes in the training set (Appendix A), a total of 11 genes associated with the OS were identified (Figure 2a). Figure 2b demonstrates the changing trajectories for each independent variable in the LASSO regression analysis. The independent variable coefficients progressively increased to zero with a gradual increase in the lambda value. Using a 10-fold cross-validation method, we constructed the risk model. Figure 2c demonstrates the confidence interval (CI) in correlation with different lambda values. It is shown that the model was most stable if the number of genes is ten. We excluded *CASP8*, as its inclusion would make the model most unstable. Figure 2d shows the regression coefficients of genes used in our model. Based on this model with ten necroptosis-associated genes, RSs were calculated using the following formula:RS = ΣCoef(i)*Exp(i)(Coef: regression coefficient, Exp: gene expression).

Among the ten genes, *ID1*, *PLK1*, *MLKL* and *HSPA4* were identified as risk factors, whereas *IDH2*, *RIPK3*, *FASLG*, *BRAF*, *ITPK1* and *TNFSF10* were identified as protective factors in terms of OS. Following this, the RS of each sample in the training set was calculated using the abovementioned formula to determine the impact of RSs on patients’ OS.

Samples of the training set were divided into a high-risk group (HRG) and a low-risk group (LRG) with the median RS as the cutoff. A KM survival analysis revealed that the OS was shorter among patients in the HRG than among those in the LRG (Figure 2e). According to the ROC analysis, the area under the curve (AUC) values for the prediction of the 1-, 2- and 3-year OS were 0.824, 0.862 and 0.785 (Figure 2f).

### 3.3. Robustness of RS in Different Cohorts

To determine the stability of the developed risk model, the abovementioned formula was used for the calculation of the RS of each sample in the GSE140082 testing set, the whole GSE140082 set and the TCGA-OV dataset. All sets were divided into an HRG and LRG based on their median RSs. A KM analysis revealed that the OS was significantly shorter among patients in the HRG than among patients in the LRG in all three datasets (Appendix A).

In the ROC analysis, the AUC values for the prediction of the 1-, 2- and 3-year OS via our RS vary only slightly by 0,7 in the three datasets: the GSE140082 testing set, whole GSE140082 set and TCGA-OV set (Appendix A).

### 3.4. Risk Model Can Predict the Prognosis of Patients with Different Clinical Characteristics

A subgroup analysis was performed in the entire GSE140082 dataset to examine the applicability of our risk model in a prognostic analysis of EOC patients with different characteristics. As shown in the KM curves in Figure 3a–f, the prognosis was significantly worse in the HRG than in the LRG in all subgroups of age, FIGO stage and grading.

Furthermore, the RS values of our clinical subgroups were compared. It showed that the RS values of patients aged >60 years were significantly higher than those of patients aged ≤60 years (Figure 3g) as well as the RS values in FIGO stage IV compared to stage III (Figure 3h). Contrasting this, grading showed no significant difference (Figure 3i). Additionally, the RS values were significantly higher in cluster C than in clusters A and B (Figure 3j).

These results were verified in the TCGA-OV dataset, and the results were consistent with those obtained via data analysis of the whole GSE140082 dataset (Appendix A). Therefore, the established risk model can effectively predict the prognosis of patients with different clinical characteristics.

### 3.5. RS Can Be an Independent Risk Factor for the Prognosis of EOC Patients

A univariate Cox regression analysis of the GSE140082 dataset revealed that the RS value was significantly correlated with the OS (hazard ratio (HR) = 4.41, 95% CI = 2.71–7.18, *p* < 0.001). After adjusting for different confounding factors, we added the multivariate Cox regression analysis, which revealed that the RS was also an independent predictor of OS (HR = 3.47, 95% CI = 2.10–5.74, *p* < 0.001) (Figure 4a,b).

Subsequently, a nomogram was constructed by integrating RSs and age to predict the survival of patients with EOC, as the multivariate Cox analysis revealed both of these factors as independent (Figure 4c). The calibration curve in Figure 4d demonstrated the accurate prediction of 1-, 2- and 3-year OS by the nomogram. Additionally, ROC curves showed, compared with the use of a single prognostic factor, superior AUC values for predicting 1-, 2- and 3-year OS (Figure 4e–g). An analysis of TCGA-OV data using the same method revealed consistent results (Appendix A).

### 3.6. Immune Cell Infiltration Was Lower in the HRG than in the LRG

To determine the correlation between RSs and the TIME, a single-sample GSEA was used to evaluate the infiltration levels of 23 types of immune cells in the GSE140082 dataset (Figure 5a). Our results showed varying infiltration levels, but the majority of immune cell types were less represented in the HRG.

Further, the ESTIMATE algorithm was used to calculate immune, ESTIMATE, stromal and tumor purity scores. Immune and ESTIMATE scores were significantly lower in the HRG (Figure 5b,c). Contrastingly, stromal and tumor purity scores were significantly lower in the LRG (Figure 5d,e).

### 3.7. Response to Immunotherapy Was Better in LRG than in HRG

In this study, the immunotherapy cohort IMvigor210 was used to examine whether the necroptosis-associated genes could potentially predict the benefits of an immunotherapeutic approach: a PD-L1 blockade. We focused on this type of immune checkpoint blockade as genes and corresponding pathways incorporated in our RS seemed to be most likely associated with PD-L1 expression and its downstream effects compared to other immune checkpoint blockades.

A KM analysis revealed that the prognosis of urothelial carcinoma patients from this cohort was worse in the HRG than in the LRG (Figure 6a). The proportion of patients who responded to the PD-L1 blockade (CR/PR) was significantly higher in the LRG than in the HRG (28.5% versus 17.0%) (Figure 6b). A violin plot revealed that RSs were significantly higher in non-responders (SD/PD) than in responders (CR/PR) (Figure 6c).

To verify the therapeutic response, RSs, neoantigen (NEO) data and tumor mutation burden (TMB) data of the IMvigor210 cohort were integrated using logistic regression. The combined AUC was 77.2%, which was higher than the single values of the RS (AUC = 0.703), TMB (AUC = 0.671) and NEO (AUC = 0.715) (Figure 6d).

To understand the distribution of immune cells, 22 types of them were calculated using the CIBERSORT algorithm for the IMvigor210 cohort. The infiltration levels of plasma cells, dendritic cells, activated mast cells and neutrophils were significantly higher in the HRG than in the LRG. In contrast, the infiltration levels of gamma delta T cells and M0/M1-type macrophages were significantly lower in the HRG than in the LRG (Figure 6e). Additionally, using the MCP counter to evaluate the infiltration of ten types of immune cells in the whole IMvigor210 cohort, the RSs were found to have a negative correlation with the scores of a majority of immune cell types (Figure 6f).

### 3.8. High Nuclear Expression of RIPK3 Is Associated with Significantly Longer Progression-Free Survival (PFS) and OS in an EOC Patient Cohort

Table 3 shows the patient characteristics, including histological subtypes, grading and FIGO staging classification of specimens in a dataset comprising 155 patients with EOC. The median OS of this cohort was 3.6 years (95% CI = 2.0–5.3 years).

Nuclear staining of RIPK3 was performed in samples obtained from 144 patients (RIPK3 staining was not successful in the remaining 11 patients owing to technical reasons). With the *survminer* package, we calculated the optimal density gradient threshold of RIPK3 expression, dividing the samples in two groups: high- and low-RIPK3-expression. For the OS as a prognostic indicator, the corresponding optimal threshold of RIPK3 expression was IRS = 7 (Figure 7a) and for the PFS IRS = 6.67 (Figure 7c).

OS was longer in the high-RIPK3-expression group than in the low-RIPK3-expression group, but without significance (*p* = 0.05) (Figure 7b). In parallel, PFS was significantly longer in the high-RIPK3-expression group than in the low-RIPK3-expression group (*p* < 0.004) (Figure 7d). Representative staining images with scores are shown in Figure 7e–h.

## 4. Discussion

EOC is the fifth most common cause of cancer-related deaths among women worldwide. As early-stage EOC has no evident symptoms, patients are frequently diagnosed in an advanced stage. In total, 80% of patients with EOC develop metastases [9,10]. Nowadays, EOC is usually treated with cytoreductive surgery followed by platinum-based chemotherapy in combination with bevacizumab and/or poly (ADP-ribose) polymerase (PARP) inhibitors. Despite recent improvements through the introduction of PARP inhibitors, the overall prognosis remains poor. The predominant mechanism of action of most systemic therapeutics so far is the inhibition of tumor growth by inducing apoptosis [25]. However, tumor cells are often resistant to undergoing apoptosis and switch over to necroptosis [11]. Studies in EOC and other entities have reported that tumor cells resistant to apoptosis can be sensitive to necroptosis [11,26,27], suggesting that necroptosis-associated genes can be used as targets to predict the prognosis of tumors.

In this study, EOC FIGO stage III/IV was initially categorized into three clusters based on the expression of 66 necroptosis-associated genes. The clusters showed significant differences in OS, representing several molecular subtypes of EOC, and are necessary for the construction of the gene signature.

From the necroptosis-associated genes, relevant to the prognosis, a 10-gene signature based on *IDH2*, *RIPK3*, *FASLG*, *BRAF*, *ITPK1*, *TNFSF10*, *ID1*, *PLK1*, *MLKL* and *HSPA4* was constructed. With an RS built from the expression of these 10 genes in tumor samples, patients’ prognosis can be estimated using a nomogram. This score showed strong predictive ability in different datasets and universality among patients with different clinical characteristics. Among these genes, high expressions of *ID1*, *PLK1*, *MLKL* and *HSPA4* are risk factors, whereas high expressions of *IDH2*, *RIPK3*, *FASLG*, *BRAF*, *ITPK1* and *TNFSF10* are protective factors. For some of these genes, a correlation with necroptosis in malignant tumors is already known.

Isocitrate dehydrogenase 2, coded by *IDH2,* is located in mitochondria and is essential for maintaining their redox homeostasis [28,29,30]. Many cancer studies already focused on *IDH1* and *IDH2* mutations. In lower-grade glioma, for example, mutant *IDH* patients were shown to have a better OS and PFS. Interestingly, in mutant as well as non-mutant gliomas, RIPK3 expression can additionally stratify the risk of patients: high RIPK3 expression is correlated with a worse prognosis contrasting to most other entities [8].

RIPK3 forms the necrosome with RIPK1 and MLKL, consequently induces necroptosis and herewith logically determines its position in our gene signature [12,13,31,32,33]. Besides necroptosis, it is involved in at least four other pathways including caspase-mediated apoptosis and NF-κB-mediated cell proliferation. Interestingly, both down-regulation or over-expression in malignant tumors are possible. In our model, over-expression is associated with a longer OS and PFS—herewith, *RIPK3* belongs to the group of protective factors in our RS. This is in line with pancreatic ductal adenocarcinoma and esophageal cancer [7] or cervical cancer, for example [34]. But, the reproducibility of many studies around this topic seems to be poor, and as for other entities like colon cancer or breast cancer, contrasting data exist on the role of RIPK3 protein [7]. The pro- or anti-carcinogenesis functions of *RIPK3* signaling are dependent on a complex balance of cytokines and chemocytokines [2,7]. As its gene product is one of the main players in necroptosis, we choose this to investigate its expression in more detail. Firstly, we found it overexpressed in EOC in a pan-cancer analysis, supporting its role in this entity compared to others. Furthermore, in an EOC patient cohort from our department, we detected a higher nuclear expression of RIPK3 to be correlated significantly with prolonged PFS and to be tending toward a longer OS. These in vivo results go along with our established gene signature. However, it should be noted that this cohort dates back to a time before the PARP inhibitor era, which puts their lifetime data into perspective.

*FASLG*, coding for the natural ligand of Fas, is a transmembrane protein that belongs to the tumor necrosis factor family. In polymyositis, the death of myofibroblasts is moderated by Fas/Fas ligand-dependent necroptosis, and the inhibition of necroptosis therefore can improve muscle weakness caused by myositis [35]. In our study, the high expression of *FASLG* as a driver of necroptosis induction is a protective factor. *FASLG* has already been used in other signatures from necroptosis-associated genes to predict the prognosis of other cancers like renal clear cell carcinoma [36] or skin melanoma [37].

*BRAF* is an important regulator of cell survival, protein synthesis, cell growth and proliferation [38]. The serine/threonine-protein kinase domain of B-Raf has a high sequence similarity with that of RIPK1 and RIPK3 [39]. Therefore, B-Raf inhibitors can also inhibit the kinase activity of RIPK1 and RIPK3 and herewith limit programmed cell death. Additionally, a decrease in the mRNA expression of RIPK3 during tumor growth in patients with colorectal cancer, gastric cancer and EOC is driven by *BRAF* overactivation according to Najafov et al. [40]. This correlation seems to be contradicting to our gene signature, where both *RIPK3* and *BRAF* act as protective factors, which would be supported by the sequence similarity mentioned [39]. An explanation for this could be that Najafov et al. found the inhibition of RIPK3 expression especially in the subgroup of samples with oncogenic mutated *BRAF* overactivation [40], whereas we did not focus on that mutation status.

As an important regulatory enzyme of the phosphatidylinositol signaling pathway, inositol-tetrakisphosphate 1-kinase (ITPK1) is crucial for the activation of viral infection-induced necroptosis [41]. It is in line with our results, where the over-expression of *ITPK1* and herewith the putative activation of necroptosis was found to be a protective factor. This is supported by the fact that through *ITPK1* mutations, leading to lower activity, the downstream necroptosis signal can be inhibited [42].

The mitotic polo-like kinase 1 (PLK1), a risk factor in our signature, is already known to be upregulated in androgen-insensitive prostate cancer. Small-molecule inhibitors of PLK1 can lead to the necroptosis of prostate cancer cells [43]. Further, in the G2 and M phase, PLK1-mediated phosphorylation of the S369 site of RIPK3 enables it to trigger apoptosis within ribosomes and necroptosis outside ribosomes [44].

Regardless of the shown data available for some of the individual genes, it is important to mention that the relevance of our gene signature lies in the interaction of the individual gene expressions.

Studies have shown that the necroptosis of tumor cells can influence the TIME by supporting the infiltration of immune-related cells such as M2 macrophages or myeloid-derived suppressor cells (MDSCs) [45], whereas the necroptosis of endothelial cells in the TIME may promote the invasion and migration capability of tumor cells [46,47]. A mouse model, used for the investigation of pancreatic cancer, showed that the depletion of RIPK3 led to the formation of a suppressive TIME, which promoted tumorigenesis [45]. Herewith, a strong association between RIPK3-dependent necroptosis and the TIME was confirmed, which led us to explore the relationship between our RS and clustering to different immune scores.

We evaluated the immune, stromal and ESTIMATE scores and the infiltration of ten types of immune cells via the MCP counter in the different clusters A, B and C based on necroptosis-associated genes and representing molecular subtypes of EOC. Herewith, differences in survival, potentially influenced by differences in TIME characteristics, can be shown. The three scores as well as almost all levels of different immune cell types were lower in cluster C than in clusters A and B. Since cluster C showed the worst clinical prognosis, it may have lacked from immune cells, leading to less elimination of antigens, higher tumor cell infiltration and reduced natural immune defense towards EOC cells. The correlation of a shorter OS with low immune infiltration is in line with results from other studies [48]. This was also proven through our RS, as immune cell infiltration and immune as well as ESTIMATE scores were found to be lower in the HRG than in the LRG.

Previous studies have reported that the presence of various immune cells in tumors may also result in a so-called immune-rejection phenotype—however, these cells mostly remain in the matrix surrounding the tumor cell nest instead of infiltrating it [46]. Further, T cells are thought to be inhibited via matrix activation in the TIME [47]. In this study, the matrix score was highest in cluster A, while the prognosis of cluster A was in between the two other clusters. We speculate that matrix activation in cluster A prevents the antitumor effects of immune cells herein.

To estimate prognostic consequences from this data, the potential response to a PD-L1 blockade treatment was evaluated. Contrasting with endometrial and cervical cancer, EOC studies on immune checkpoint inhibitors revealed disappointing data for several years. Studies on the PD-L1 blockers avelumab [49] or atezolizumab [50,51] did not reach their primary endpoint of PFS. But in 2023, data from Harter et al. firstly showed a benefit in PFS after the addition of the programmed cell death protein 1 (PD-1) blocker durvalumab to chemotherapy, bevacizumab and olaparib [52]. As the immune infiltration levels already suggested, in our study, the response to a blockade of the interaction between PD-1 and PD-L1 was worse in the HRG than in the LRG. This statement is limited by the fact that we used a cohort of patients with urothelial carcinomas, known to have a less immunosuppressive TIME, to predict the effect of immunotherapy in the absence of a sufficient EOC cohort. Furthermore, according to the PD-1 data from Harter et al., it is necessary to examine this type of inhibition separately [52], since we used a cohort with a PD-L1 blockade. Nevertheless, in anticipation of future study results on the implication of immunotherapy in EOC treatment [53,54,55], risk models such as ours might be able to select patients who are eligible for this therapy.

A limitation of our study is the retrospective analysis of data, which needs to be consolidated prospectively to implement their significance. Further, the in silico correlations had to be proven in vivo and/or in vitro to determine the mechanism of action of necroptosis-associated genes during the onset and progression of EOC. Interestingly, Qin et al. also published a signature for EOC made of necroptosis-associated genes. In contrast to our study, they only used *CXCL10*, *RELB* and *CASP3* to build an RS and examined its association with the TIME [56]. Both studies emphasize herewith the importance of necroptosis-related pathways in the prognosis of EOC and TIME regulation. However, it should be noted that for a generalized application, the use of a gene signature generated from a broader pool of genes is obvious. As there are a couple of different signatures for advanced EOC [56,57,58,59], future research should aim for one signature which includes this wide spectrum of biochemical pathways, especially in a time period where next-generation sequencing is no longer a relevant economic problem.

## 5. Conclusions

We created a 10-gene signature from the expression of necroptosis-associated genes in EOC. Herewith, an RS for EOC patients can be calculated to predict their prognosis. Furthermore, this RS could potentially be useful to predict the response to a PD-L1 blockade treatment in selected cancer patients. We thus contribute to the development of a broader gene signature for the risk stratification of EOC and provide an idea of how the use of immunotherapy can potentially be guided.

## Figures and Tables

**Figure 1 cancers-17-00271-f001:**
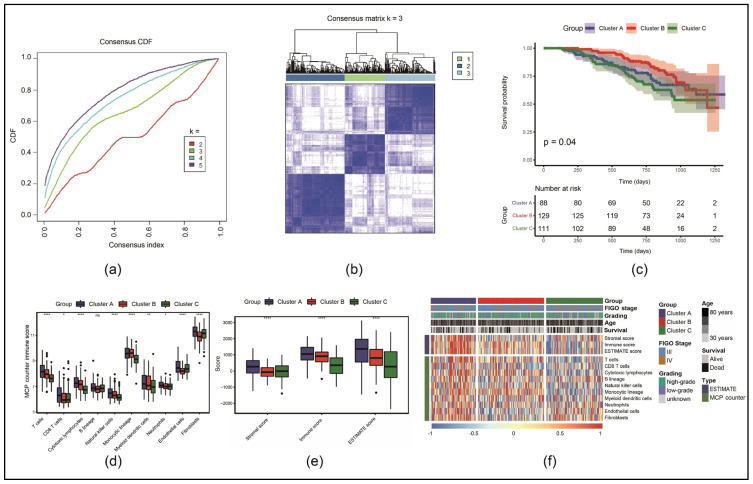
Identification of three necroptosis molecular clusters: (**a**) cumulative distribution function (CDF) curve of consistent clustering; (**b**) clustering results when the number of categories is three; (**c**) Kaplan–Meier (KM) survival curves; (**d**) comparison of Microenvironment Cell Populations (MCPs) counter immune scores (ns: *p* ≥ 0.05, * *p* < 0.05, ** *p* < 0.01, **** *p* < 0.0001); (**e**) comparison of Estimation of STromal and Immune cells in MAlignant Tumor tissues using Expression data (ESTIMATE), stromal and immune score; (**f**) clustering heatmaps of immune cell scores across different clinical features and groupings.

**Figure 2 cancers-17-00271-f002:**
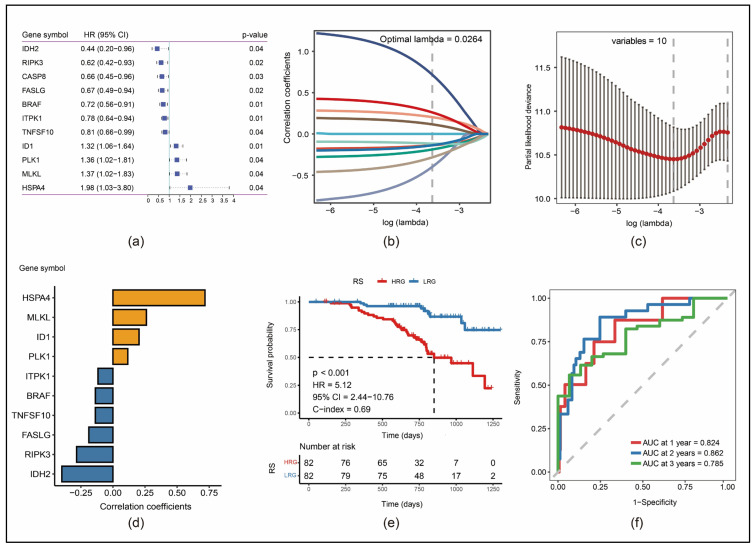
Construction of a multiple gene signature: (**a**) hazard ratio (HR) with the 95% confidence interval (CI) and the resulting *p*-value of the 11 prognosis-associated necroptosis genes identified with univariate Cox analysis; (**b**) change trajectories of each independent gene variable; (**c**) CIs in relation to each lambda; (**d**) correlation coefficients for each of the remaining ten genes; (**e**) KM survival curves for high-risk group (HRG) and low-risk group (LRG) regarding the risk score (RS) in the training set; (**f**) time-dependent receiver operating characteristic (ROC) curves in the training set and their resulting areas under the curve (AUC).

**Figure 3 cancers-17-00271-f003:**
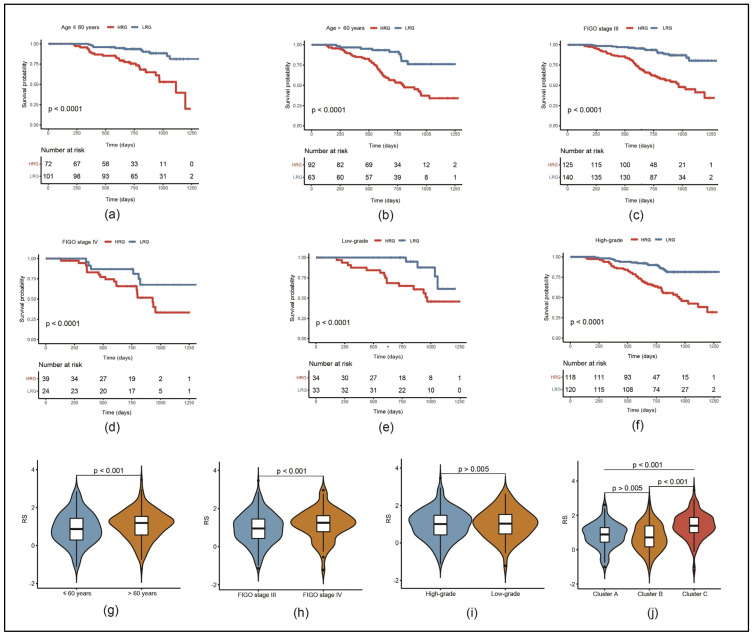
Analysis of clinical subgroup prognoses based on our RS in the GSE140082 dataset: (**a**–**f**) KM curves comparing HRG and LRG in different clinical subgroups regarding age, FIGO stage and grading; (**g**–**j**) comparison of RSs in contrasting subgroups regarding the same parameters and our molecular clusters A, B and C.

**Figure 4 cancers-17-00271-f004:**
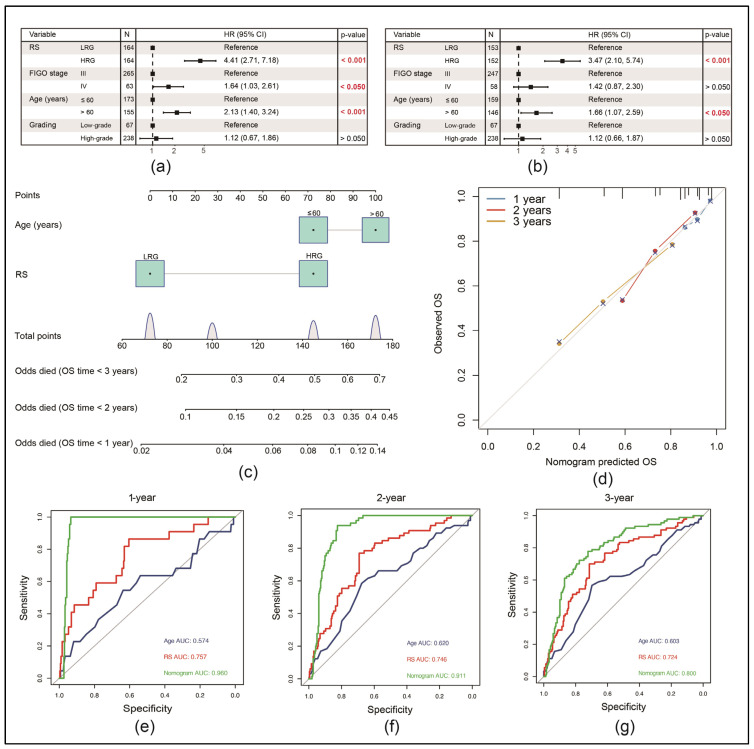
Proving the RS as an independent risk factor for the prognosis of EOC patients in the GSE140082 dataset: (**a**) univariate Cox analysis with significant *p*-values marked red; (**b**) multivariate Cox analysis with significant *p*-values marked red; (**c**) construction of a nomogram using significant variables from the multivariate Cox analysis; (**d**) calibration curve comparing the observed 1-, 2- and 3-year overall survival (OS) with the predicted OS from the nomogram; (**e**–**g**) ROC curves comparing the nomogram with the used single variables for the prediction of the 1-, 2- and 3-year OS.

**Figure 5 cancers-17-00271-f005:**
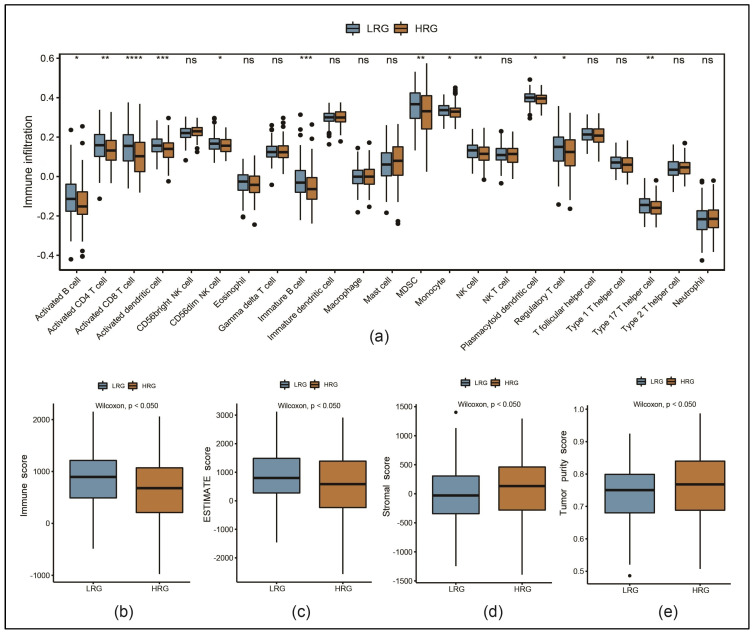
Immune cell infiltration of the tumor immune microenvironment (TIME) in the GSE140082 dataset: (**a**) boxplots representing the infiltration levels of 23 types of immune cells comparing HRG and LRG (ns: *p* ≥ 0.05, * *p* < 0.05, ** *p* < 0.01, *** *p* < 0.001, **** *p* < 0.0001, NK: natural killer, MDSCs: myeloid-derived suppressor cells); (**b**–**e**) comparing immune, ESTIMATE, stromal and tumor purity scores between LRG and HRG.

**Figure 6 cancers-17-00271-f006:**
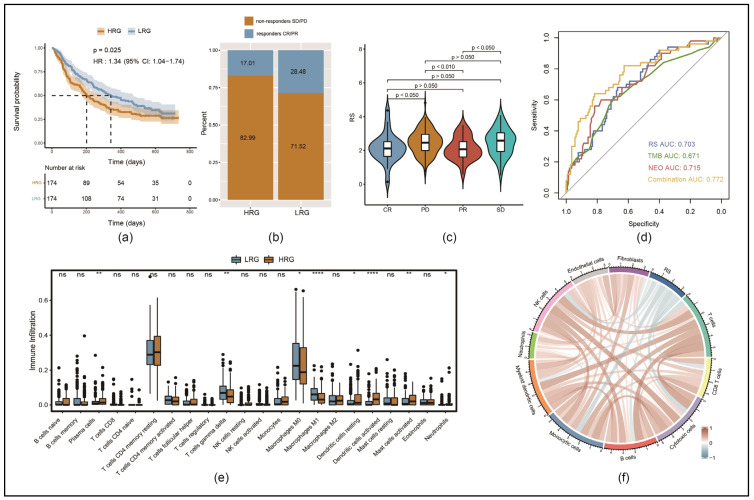
Response to programmed death ligand 1 (PD-L1) treatment regarding the RS in the immunotherapy cohort IMvigor210: (**a**) KM survival analysis comparing LRG and HRG; (**b**) comparing the proportion of immunotherapy responders (partial response (PR) and complete response (CR)) versus non-responders (progressive disease (PD) and stable disease (SD)) in both groups; (**c**) comparison of differences in RSs according to the efficacy of PD-L1 blockade, resulting in CR, PR, SD or PD; (**d**) ROC curves comparing the predictive accuracy of the RS, the tumor mutation burden (TMB), the neoantigen (NEO) or the combination of all three of them; (**e**) boxplots representing the infiltration levels of 22 types of immune cells comparing HRG and LRG (ns: *p* ≥ 0.05, * *p* < 0.05, ** *p* < 0.01, **** *p* < 0.0001); (**f**) correlation analysis of RS and different types of immune cells.

**Figure 7 cancers-17-00271-f007:**
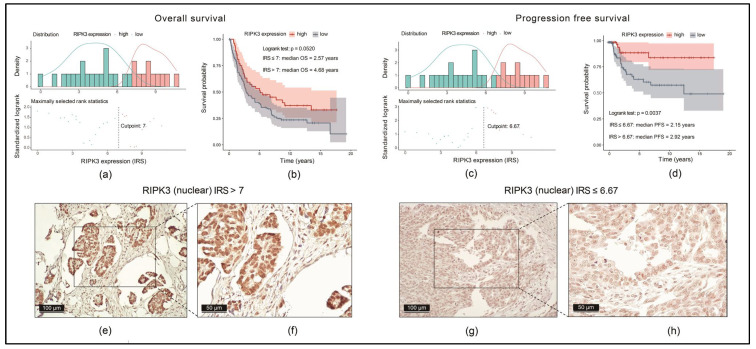
Immunohistochemical nuclear expression of RIPK3 in an EOC cohort and its correlation with OS and PFS: (**a**,**c**) optimal density gradient thresholds of RIPK3 expression calculated as immunoreactivity score (IRS); (**b**,**d**) KM analysis for OS and progression-free survival (PFS) high- and low-RIPK3-expression; (**e**–**h**) characteristic stainings of RIPK3 with different IRSs in two scales (100 µm and 50 µm).

**Table 1 cancers-17-00271-t001:** Clinical features of the gene expression datasets.

Clinical Features	TCGA-OV (*n* = 347)	GSE140082 (*n* = 328)
**Age**		
≤60 years	192 (55.3%)	173 (52.7%)
>60 years	155 (44.7%)	155 (47.3%)
**FIGO stage**		
III	290 (83.6%)	265 (80.8%)
IV	57 (16.4%)	63 (19.2%)
**Grading**		
Low	36 (10.4%)	67 (20.4%)
High	303 (87.3%)	238 (72.6%)
Unknown	8 (2.3%)	23 (7.0%)
**3-year survival**		
Alive	125 (36.0%)	92 (28.0%)
Dead	222 (64.0%)	236 (72.0%)

Age, Fédération Internationale de Gynécologie et d’Obstétrique (FIGO) stage, grading and 3-year survival in the epithelial ovarian cancer (EOC) cohort of The Cancer Genome Atlas (TCGA), called TCGA-OV, and the Gene Expression Omnibus (GEO) series 140082 (GSE140082) cohort.

**Table 2 cancers-17-00271-t002:** GSE140082 grouping information.

	GSE140082 Training (*n* = 164)	GSE140082 Testing (*n* = 164)	*p*-Value
**Age**			
≤60 years	92 (56.1%)	81 (49.4%)	0.27
>60 years	72 (43.9%)	83 (50.6%)	
**FIGO stage**			
III	128 (78.0%)	137 (83.5%)	0.26
IV	36 (22.0%)	27 (16.5%)	
**Grading**			
Low	30 (18.3%)	37 (22.6%)	0.33
High	124 (75.6%)	114 (69.5%)	
Unknown	10 (6.1%)	13 (7.9%)	
**3-year survival**			
Alive	42 (25.6%)	50 (30.5%)	0.39
Dead	122 (74.4%)	114 (69.5%)	

Age, FIGO stage, grading and 3-year survival in both GSE140082 training and testing set.

**Table 3 cancers-17-00271-t003:** Clinical EOC patient cohort information.

		Number of Patients	Percentage
**Age**			
≤60 years		84	54.2
>60 years		71	45.8
**FIGO stage**			
I		34	21.9
II		10	6.5
III		103	66.5
IV		3	1.9
Unknown		5	3.2
**Subtype and grading**			
**Serous**	Low grade	23	14.8
	High grade	80	51.6
	Unknown	6	3.9
	Total	109	70.3
**Endometrioid**	G1	6	3.9
	G2	5	3.2
	G3	8	5.2
	Unknown	2	1.3
	Total	21	13.6
**Mucinous**	G1	6	3.9
	G2	6	3.9
	G3	0	0.0
	Unknown	1	0.6
	Total	13	8.4
**Clear-cell**	G1	2	1.3
	G2	2	1.3
	G3	5	3.2
	Unknown	3	1.9
	Total	12	7.7

Age, FIGO stage, subtype and grading.

## Data Availability

Publicly available datasets were analyzed in this study and can be found here: University of California Santa Cruz Xena (https://gdc.xenahubs.net [accessed on 10 December 2024]) and the GEO (https://www.ncbi.nlm.nih.gov/geo/ [accessed on 10 December 2024]). Data generated by the authors are shown in this paper or in the Appendix A. Further data are available upon request from the corresponding author if they are not shown somewhere else.

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
