# Peer review of "Necroptosis-Related Gene Signature Predicts Prognosis in Patients with Advanced Ovarian Cancer"

_cancers, 2025, doi:10.3390/cancers17020271_

Round 1
Reviewer 1 Report
Comments and Suggestions for Authors
The study addresses an important clinical challenge by developing a risk score (RS) based on necroptosis-associated genes to predict prognosis and response to PD-L1 blockade in epithelial ovarian cancer (EOC), representing a significant contribution to the field. While the manuscript presents a novel and promising tool for prognosis prediction in EOC, revisions are necessary to address identified weaknesses, improve clarity, and enhance overall quality. With these improvements, the study has the potential to make a substantial impact in the field of oncology research.
Epithelial ovarian cancer (EOC) is the fifth leading cause of cancer-related deaths in women. However, the authors provide conflicting statements, referring to it as the second most common in the introduction but citing it as the fifth most common cause of cancer-related deaths among women in the discussion section.
For Figure 1, the classification criteria for Cluster A, B, and C need to be clearly explained. Providing a detailed description of the parameters or characteristics used to define each cluster would ensure better understanding and interpretation of the figure.
Figure 3 needs to be presented in higher resolution with clearer and more legible labeling to ensure all details can be easily read and understood. The overall resolution of the figures is suboptimal, and some figure labels are difficult to read.
The reason for specifically analyzing the response rate of PD-L1 blockade with RS (Risk Score) rather than other immune checkpoint blockades (such as anti-PD1, anti-CTLA4, or anti-VISTA) should be clearly justified.
The manuscript lacks in-depth discussion on the limitations of using a cohort of urothelial carcinoma patients (IMvigor210) to predict immunotherapy responses in EOC due to the absence of relevant EOC datasets.
Reviewer 2 Report
Comments and Suggestions for Authors
Authors have prepared a very interesting paper giving more insight into ovarian cancer, which still remains a challenge with no significant treatment improvement over the last 30 years. Authors have also nicely summed up the limitations of their study and the need for further in vitro and in vivo studies. For improving the paper, i would suggest:
1. A similar paper has already been published early this year, I suggest to compare results and cite: https://doi.org/10.1038/s41598-024-61849-y
2. Discussion is very long, please shorten.
3. Please include mAB information (cat. no., lot no.) for anti-RIPK3 antibody and include all other relevant info regarding the experiments ( other chemicals used, times of incubation). Experiments cant be repeated from the current description.
4. What were the positive and negative controls for IHC stainings? How was the staining validated? Were controls included on every microarray prepared?
